# The OPRM1 gene and interactions with the 5-HT1a gene regulate conditioned pain modulation in fibromyalgia patients and healthy controls

**Jeanette Tour**[1,2,3]*, **Angelica Sandström**[1,2], **Diana Kadetoff**[1,4], **Martin Schalling**[5], **Eva Kosek**[1,2,6]

1 Department of Clinical Neuroscience, Karolinska Institutet, Stockholm, Sweden, 2 Department of Neuroradiology, Karolinska University Hospital, Stockholm, Sweden, 3 Department of Oncology, Blekinge Hospital, Karlskrona, Sweden, 4 Stockholm Spine Center, Löwenströmska Hospital, Upplands Väsby, Sweden, 5 Department of Molecular Medicine and Surgery, Karolinska Institutet, Center for Molecular Medicine, Karolinska University Hospital, Stockholm, Sweden, 6 Department of Surgical Sciences, Uppsala University, Uppsala, Sweden

* Jeanette.tour@ki.se

**Data Availability Statement:** Data cannot be shared publicly because of restrictions from the

## Abstract

Fibromyalgia (FM) patients have dysfunctional endogenous pain modulation, where opioid and serotonergic signaling is implicated. The aim of this study was to investigate whether genetic variants in the genes coding for major structures in the opioid and serotonergic systems can affect pain modulation in FM patients and healthy controls (HC). Conditioned pain modulation (CPM), evaluating the effects of ischemic pain on pressure pain sensitivity, was performed in 82 FM patients and 43 HC. All subjects were genotyped for relevant functional polymorphisms in the genes coding for the μ-opioid receptor (OPRM1, rs1799971), the serotonin transporter (5-HTT, 5-HTTLPR/rs25531) and the serotonin 1a receptor (5-HT1a, rs6295). Results showed the OPRM1 G-allele was associated with decreased CPM. A significant gene-to-gene interaction was found between the OPRM1 and the 5-HT1a gene. Reduced CPM scores were seen particularly in individuals with the OPRM1 G*/5-HT1a CC genotype, indicating that the 5-HT1a CC genotype seems to have an inhibiting effect on CPM if an individual has the OPRM1 G-genotype. Thus, regardless of pain phenotype, the OPRM1 G-allele independently as well as with an interaction with the 5-HT1a gene influenced pain modulation. FM patients had lower CPM than HC but no group differences were found regarding the genetic effects on CPM, indicating that the results reflect more general mechanisms influencing pain modulatory processes rather than underlying the dysfunction of CPM in FM. In conclusion, a genetic variant known to alter the expression of, and binding to, the my-opioid receptor reduced a subject's ability to activate descending pain inhibition. Also, the results suggest a genetically inferred gene-to-gene interaction between the main opioid receptor and a serotonergic structure essential for 5-HT transmission to modulate pain inhibition. The results in this study highlight the importance of studying joint synergistic and antagonistic effects of neurotransmittor systems in regard to pain modulation.

Swedish Ethical Review Authority board. Data can be available after approval from the Swedish Ethics Committee (contact via https://etikprovningsmyndigheten.se/en/) for researchers who meet the criteria for access to confidential data.

**Funding:** The research leading to these results has received funding from a generous donation from Leif Lundblad and family, the European Union Seventh Framework Programme (FP7/2007-2013) under grant agreement No 602919, the Swedish Research Council, Stockholm County Council 20190039, CSTP grant (Tour) from Karolinska Institutet, a Swedish patient organization for fibromyalgia, "Fibromyalgiforbundet" and support from Vetenskapliga Radet, Blekinge Hospital.

**Competing interests:** The authors have declared that no competing interests exist.

## Introduction

Fibromyalgia (FM) is characterized by generalized long-term pain, often along with non-restorative sleep, fatigue and cognitive impairment like memory loss [1–3]. The pain in FM is regarded as nociplastic [4–6], as FM patients fulfill the nociplastic pain criteria [7] based on the characteristic pain hypersensitivity [8,9]. Furthermore, dysfunctional descending pain inhibition, i.e., reduced exercise-induced hypoalgesia (EIH) [10,11] and conditioning pain modulation (CPM) [12,13] has been reported in FM patients.

Pain modulatory processes involve several neuromodulators including opioids and serotonin (5-HT). In healthy subjects, endogenous opioid as well as 5-HT signaling has been implicated to affect CPM [14,15]. In FM patients, indirect evidence of increased opioid- and reduced 5-HT metabolism has been described [16,17]. There is also evidence of interactions between opioid and serotonergic signaling. Animal studies have shown that 5-HT1a agonists can prevent and reverse opioid-induced hyperalgesia [18]. Also, the efficacy of an opioid agonist was increased in individual carriers of a serotonergic genotype proposed to reduce 5-HT signaling [19]. We recently showed that, by genetic association, increased endogenous opioid tone along with lower serotonergic signaling produced more profound EIH [11], a paradigm similar to but not identical with CPM [20,21]. To test the generalizability of this finding, we here study a different paradigm of endogenous pain inhibition, i.e. CPM, and in a different cohort of FM patients and healthy controls (HC).

The aim of this study was to assess the influence of specific pain related genetic variants on CPM. More specifically, the single nucleotide polymorphism (SNP) *rs1799971* in the OPRM1 gene, regulating the expression, availability, as well as the activation of the μ-opioid receptors (MOR) [22]. Individuals with the G-genotype (A/G or G/G) are proposed to have increased receptor affinity for β-endorphins [23], thus increased endogenous opioid efficacy compared with the homozygous AA-genotype, however, there are conflicting results [24]. A meta-analysis concluded recently that individuals with the OPRM1 G-genotype have reduced analgesic efficacy of opioid drugs [25]. Also, this variant has been shown to influence cerebral processing during evoked pain regardless of pain phenotype [26] and has been linked to reduced cerebral MOR availability [27]. Moreover, we jointly studied two functional polymorphisms (*5-HTTLPR* and *rs25531*) in the serotonin transporter (5-HTT) gene SLC6A4. Together they alter the degree of gene expression and functionally divide individuals into groups of high, intermediate or low expression of the 5-HTT [14,28], which have been associated with CPM efficacy in HC [14]. Lastly, the SNP *rs6295* in the HTR1A gene, regulates the expression of the 5-HT1a receptor [29]. Individuals with the 5-HT1a G-genotype (C/G or GG) are associated with having reduced 5-HT signaling compared with the homozygous CC-genotype [30,31] and increased sensitivity to suprathreshold pain [32].

Based on our previous findings [11] we hypothesized that individuals with OPRM1 G-genotype in combination with the 5-HTT low expression or 5-HT1a G-genotype have a more pronounced CPM, and vice versa, and that this will be seen regardless of baseline pain level, i.e., in FM patients and HC alike.

## Materials and method

### Participants

The current study forms part of a larger project, see study plan https://osf.io/8zqak [33,34]. Study participants were recruited by local news advertisement. FM patients were evaluated by an experienced physician with a standardized interview and palpation of tender points to verify both the American College of Rheumatology 1990 criteria for FM [35] and the newer 2010

criteria [36]. Inclusion criteria included female sex, right handedness and age between 20–60 years. Exclusion criteria for FM patients and HC were high blood pressure (>160/90), painful osteoarthritis, other causes of pain than FM for FM patients or any pain rated as more than 20/100 VAS (visual analog scale) for HC, other severe somatic or psychiatric disorder, addiction disorder, pregnancy, severe obesity (BMI > 35), smoking, medication with antidepressants or anticonvulsants, and inability to refrain from analgetics, anti-inflammatory drugs or hypnotics for 48 hours before participation in the study. HC were screened by telephone before their visit. A total of 117 FM patients and 45 HC were screened for participation. Subjects were excluded for not meeting the inclusion/exclusion criteria (FM n = 17, HC n = 1) and for declining participation (FM n = 14). 4 FM patients and 1 HC failed to complete the CPM assessment. Thus, data for the analysis included 82 FM patients and 43 HC. One subject was not included in the genetic analyses due to failed genotyping. All participants were Caucasian women. They were given verbal and written information and gave written informed consent in accordance with the Declaration of Helsinki. The study was reviewed and approved by the Swedish Ethical Review Authority board (permit 2014/1604-31/1) before the study began.

## Procedures

**Questionnaires.** All participants completed standardized and validated questionnaires concerning health status and quality of life. The questionnaires analyzed in the present study are Hospital Anxiety and Depression Scale (HADS), Short-Form-36 Bodily Pain Scale (SF-36 BP), and for FM patients the Fibromyalgia Impact Questionnaire (FIQ). HADS consists of 2 subscales assessing depressive (HAD-D) and anxiety (HAD-A) symptoms in nonpsychiatric patients. Each subscale comprises of 7 items with accumulated scores between 0 and 21. Cut-off scores of above 8 can be regarded as the presence of anxiety and depressive disease [37]. The SF-36 BP, a 2-item subscale within the Short Form (SF-36), was chosen to assess pain severity and pain over time (4 weeks). The raw scores are transformed into a 0 to 100 scale, where lower score indicate more pain symptoms [38]. The FIQ is a questionnaire assessing symptoms and disability related to FM. It consists of 20 items with scores ranging from 0 to 100, where higher values indicates a poorer state of health due to FM [39].

**Pressure Pain Thresholds (PPTs).** To assess pain sensitivity pressure algometry (Algometer, Somedic Sales, Sweden) was used with a probe area of 1 $cm^2$, manual pressure perpendicular to the skin was applied with an increase pressure rate of 50kPa/s. The subjects then pressed a response button at the slightest sensation of pain. PPTs were assessed bilaterally on 4 sites corresponding to the tender points of the ACR 1990 criteria for FM classification, namely musculus trapezius, lateral epicondyle of the humerus, musculus gluteus maximus, and medial proximal fat pad of knee. The assessments were conducted by two test investigators who were jointly trained to perform assessments comparably (JT,AS).

**Conditioned Pain Modulation (CPM).** CPM is a reliable method for assessing endogenous pain modulatory processes [40]. In this study, CPM was assessed with a tourniquet test for each individual. The subjects were comfortably seated upright on a stretcher with the backrest raised to approximately 50 degrees and resting their left arm on an armrest. A baseline assessment of pain sensitivity was done before the start of the tourniquet test, with 2 PPTs at the subject's right thigh (*m. quadriceps*). The left arm was raised vertically for 1 minute to drain the arm for venous blood, then a blood pressure cuff was put on the left upper arm and inflated to 200 mmHg. The subject then lifted a 1 kg weight with their left underarm extensor muscles. The subjects continuously rated their experienced pain on a VAS (0-100mm) scale. When they rated 50 mm they were told to stop lifting the weight and endure the conditioning pain. The examiner started assessing PPTs with the algometer at the subjects right (contralateral) *m. quadriceps*. PPTs were assessed

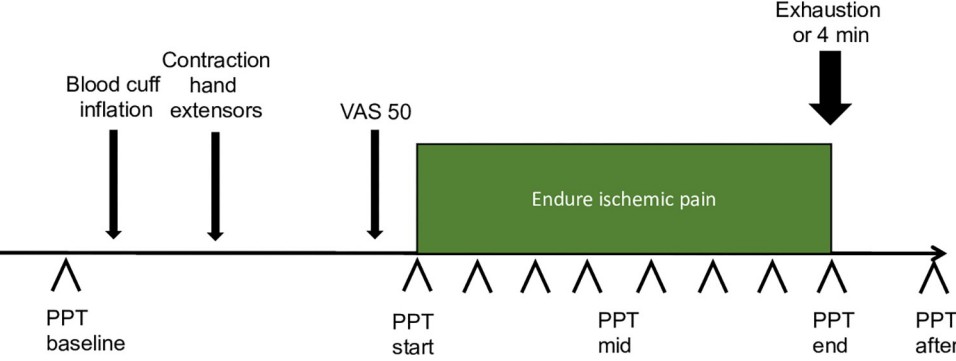

**Fig 1. Schematic view of the assessment for conditioned pain modulation.**

continuously with a minimum of 20 seconds between assessments until the subject asked to stop or for a maximum of 4 minutes. A final post CPM assessment of PPT was made 5 minutes after the end of the conditioning stimulus at *m.quadriceps* (Fig 1).

**Genotyping.** The SNPs in this study were chosen based on a priori hypotheses based on existing literature. To be consistent with previous research, the genotypes of the *rs1799971* (118A>G, N40D) of the OPRM1 gene and the C(-1019)G 5-HT1a promoter polymorphism of the HTR1A gene coding for the 5-HT1a receptor were dichotomized into major allele homozygotes and minor allele carriers [41–43]. The 5-HTT gene (SLC6A4) includes the functional polymorphism *5-HTTLPR*, which consists of an L allele and an S allele, where the S-allele is associated with reduced 5-HTT expression [44]. Moreover, the functional polymorphism *rs25531* containing an $_A$-allele and a $_G$-allele has been shown to further modulate the efficacy of *5-HTTLPR*, the minor $_G$-allele reduces gene expression to S-allele levels [28]. Given this proposed efficacy and in line with our previous studies [11,19], we examined *5-HTTLPR/rs25531* jointly, referred to as the triallelic 5-HTTLPR. This approach results in functional division of the study subjects into having high ($L_A/L_A$), intermediate ($L_A/L_G$ and $L_A/S_A$) and low ($S_A/S_A$ and $S_A/L_G$) expression of 5-HTT.

For genotyping, saliva samples (Oragene G500) were collected from all subjects. For the polymorphisms rs1799971 (OPRM1) and rs6296 (5-HT1a), genotyping was performed using TaqMan SNP genotyping assays and ABI 7900 HT instrument (Applied Biosystems (ABI), Foster City, CA). Polymerase chain reactions (PCRs), with a total volume of 5 mL, were performed in 384-well plates containing 2.5 mL Universal Master Mix (UMM) and 5 ng drieddown genomic DNA per well. The PCR amplification protocol includes 2 holds, 50˚C for 2 minutes and denaturation at 95˚C for 10 minutes, followed by 45 cycles for rs6296 and 50 cycles for rs1799971 at 92˚C for 15 seconds and 60˚C for 1 minute. For the genotyping of the triallelic 5-HTTLPR, 2 fragments, 487 bp (short) and 530 bp (long), were amplified by PCRs. Each PCR reaction contained 50 ng DNA, 0.2 mM deoxynucleotide triphosphate (dNTP), 0.4 mMof primer 17P-3F, 0.4 mM primer 17P- 3R, 0.05 mL Qiagen HotStar Polymerase, 1 M Qsolution, and finally 1x buffer. Samples were amplified on Biorad Tetrade (BIORAD, Hercules, CA) with an initial denaturation for 10 minutes at 95˚C followed by 33 cycles consisting of denaturation for 30 seconds at 95˚C, annealing for 30 seconds at 57˚C and elongation for 5 minutes at 72˚C, and finally followed by another elongation step for 5 minutes at 72˚C. Eight microliters of the PCR reactions were separated for 2 hours at 100 V by gel electrophoresis in TBE buffer on a 2.5% agarose gel containing GelRed and visualized using ultraviolet light. To determine the rs25531, 10 mL of the PCR product was digested with 0.1 mL MSP1 (New England Biolabs, Ipswich, MA) and 1 mL buffer per sample for 12 hours at 37˚C. The MSP1 restriction enzyme breaks the 59-C/CGG9 sequence that gives a fragment of 342 base pairs,

one of 127 and finally one of 62 base pairs which constitutes the LA allele, whereas the 298, 127, and 62 base pairs is the SA allele, the 173, 166, 127, and 62 base pairs for the LG allele, and finally the 166, 130, 127, and 62 for the SA allele. Fragments were run on a 4% agarose gel (3% normal agarose and 1% low melting agarose) containing GelRed initially for 15 minutes at 70 V followed by 2 more hours at 100 V. The gels were then visualized with ultraviolet light.

## Statistics

**CPM score and normalization of PPTs.**   To assess the associations between functional genetic polymorphisms and CPM, a CPM score was calculated, as the difference between the end value and the baseline value divided by the baseline value [11,14]. A positive CPM score indicates increased pain inhibition, whereas a negative CPM score indicates facilitation, i.e. increased pressure pain sensitivity during the conditioning stimulation (tournique test). In order to control for the evident interindividual differences in pain sensitivity, i.e., PPTs [45] in the assessments of CPM, the scores were normalized by dividing the PPT at each time point with the individuals first PPT at baseline (before the CPM assessment) [46]. A value over 1 indicates pain inhibition, i.e decreased sensitivity to pressure pain during conditioning stimulation. Normalized PPTs were used to analyze the temporal aspects of CPM.

**Statistical analyses.**   All analyses were made using SPSS Statistics, version 27 (SPSS Inc, Chicago, IL). Data were reported as mean with standard deviation and graphs as mean with error bars of 1 standard error of the mean. Graphs visualizing normalized PPTs were adjusted by adding a coefficient so that the baseline value corresponded to 1. Reported p-values were two-sided and $p < 0.05$ was considered statistically significant. Allelic deviation from Hardy-Weinberg equilibrium was tested with Chi-square test, and genotype frequencies were analyzed with the Fisher exact test and Chi-square test. The Shapiro–Wilk test was used to assess deviations from a normally distributed sample. To assess differences in pain sensitivity, CPM score, normalized PPTs, contraction time, number of PPT assessments and test leader differences, Students t-test and the Mann–Whitney U test was used.

The overall effects of gene-to-gene interactions were analyzed by univariate ANOVAs, with CPM score as the dependent variable, and genotypes as independent variables (OPRM1 AA/*G, 5HT1a CC/*G, 5-HTT low/intermediate/high), and age, HAD-A, and HAD-D as covariates. Post-hoc tests were performed with univariate ANOVAs and Students t-test.

The assessment of the temporal aspects of CPM were analyzed by repeated measures analysis of variance (ANOVA) with the within-subject factor TIME (normalized PPTs; baseline, start, middle, end and after 5 minutes), age as a covariate and the between subject factor GROUP (FM/HC) and GENOTYPE (OPRM1 AA/*G) in separate analyses. Post-hoc analysis was made with the Mann-Whitney U test. When the assumption of sphericity was violated the Greenhouse-Geisser correction was used. To assert possible methodological differences between the groups, the model was repeated adding the covariates for the length in time of the tourniquet test and number of PPT assessments.

Multivariate ANOVA was used to assess whether the different genotypes affected symptom parameters. The analysis was performed in FM patients with the dependent variables FIQ, HAD-D, HAD-A, average PPT, and SF-36 BP, the independent variables OPRM1, 5-HTT, and 5-HT1a, and age as a covariate.

## Results

### Participant and genotype characteristics

Patient characteristics for FM patients and HC are presented in Table 1. FM patients had significantly lower average PPT, indicating increased pain sensitivity compared to HC

**Table 1. Descriptives of study population.**

| | FM patients (N = 82) | HC (N = 43) | p-value |
|---|---|---|---|
| Age (years) | 47.2 ± 7.8 | 48.3 ± 7.7 | p = 0.47 |
| BMI (kg/m$^2$) | 26.2 ± 3.8 | 22.7 ± 2.4 | < 0.001 |
| FM duration (months) | 121 ± 88 | NA | |
| Tenderpoint count* | 17 (range 11–18) | NA | |
| Widespread Pain Index* | 15 (range 9–19) | NA | |
| Symptom Severity Score* | 10 (range 5–12) | NA | |
| Average PPT (kPa) | 150 ± 63 | 318 ± 109 | < 0.001 |
| Average VAS | 58 ± 22 | 4 ± 6 | < 0.001 |
| CPM score | -0.03 | 0.20 | p = 0.005 |
| Questionnaires: | | | |
| FIQ | 62.9 ± 16.5 | NA | |
| HAD-D | 7.2 ± 4.0 | 1.1 ± 1.5 | < 0.001 |
| HAD-A | 7.8 ± 4.2 | 3.1 ± 3.0 | < 0.001 |
| SF-36 BP | 31.3 ± 14.1 | 89.4 ± 12.5 | < 0.001 |
| Number of PPT assessments | 11.5 ± 3.2 | 11.3 ± 2.5 | p = 0.72 |
| Tourniquet time (seconds) | 192 ± 51 | 220 ± 35 | p < 0.001 |

Numbers reported as means ± SD unless otherwise indicated. FM = fibromyalgia, HC = healthy controls, NA = not applicable, BMI = body mass index, PPT = pressure pain threshold, VAS = visual analog scale, FIQ = Fibromyalgia impact questionnaire, HAD-D = Hospital anxiety and depression scale—depression, HAD-A = Hospital anxiety and depression scale—anxiety, SF-36 BP = 36-item short form–bodily pain.

*median value.

(p < 0.001) and reduced CPM score (p < 0.001) indicating a dysfunction of conditioning pain modulation. Allele frequencies were in Hardy-Weinberg equilibrium (rs1799971 X2 (1) = 3.34 p = 0.067, rs6295 X2 (1) = 0.45 p = 0.50, 5-HTTLPR X2 (1) = 0.25 p = 0.61). Genotype frequencies, presented in Table 2, were similar in both groups for all genotypes. The results did not differ between test leaders for average PPT (p = 0.17) or CPM score (p = 0.91) indicating similar methodology across the study group. The FM patient group had shorter mean tourniquets assessment time (start after rating > 50 mm VAS) with 192 seconds compared to 220 seconds in the HC group. However, given the lower PPTs in the FM group both groups had an equal number of PPT assessments.

**Table 2. Genotype frequencies of the polymorphisms _rs1799971_ (OPRM1), _rs6296_ (5-HT$_{1A}$) and the triallelic 5-HTT for the whole group, as well as fibromyalgia (FM) patients and healthy controls (HC) separately.**

| | | Whole group | | FM patients | | HC | | |
|---|---|---|---|---|---|---|---|---|
| | | N = 124 | % | N = 82 | % | N = 43 | % | |
| OPRM1 | AA | 96 | 77.4 | 60 | 74.1 | 36 | 83.7 | |
| (rs1799971) | G-carrier | 28 | 22.6 | 21 | 25.9 | 7 | 16.3 | p = 0.26 |
| 5-HT$_{1A}$ | CC | 26 | 21.0 | 16 | 19.8 | 10 | 23.3 | |
| (rs6296) | G-carrier | 98 | 79.0 | 65 | 80.2 | 33 | 76.7 | p = 0.65 |
| 5-HTT | High | 20 | 16.1 | 12 | 15.2 | 8 | 19.5 | |
| | Intermediate | 58 | 46.8 | 38 | 48.1 | 20 | 48.8 | |
| | Low | 42 | 33.9 | 29 | 36.7 | 13 | 31.7 | p = 0.78 |

Genotype frequencies did not differ between the FM and HC group (p-values). Call rate > 99% for OPRM1 and 5-HT1a and 96% for 5-HTT.

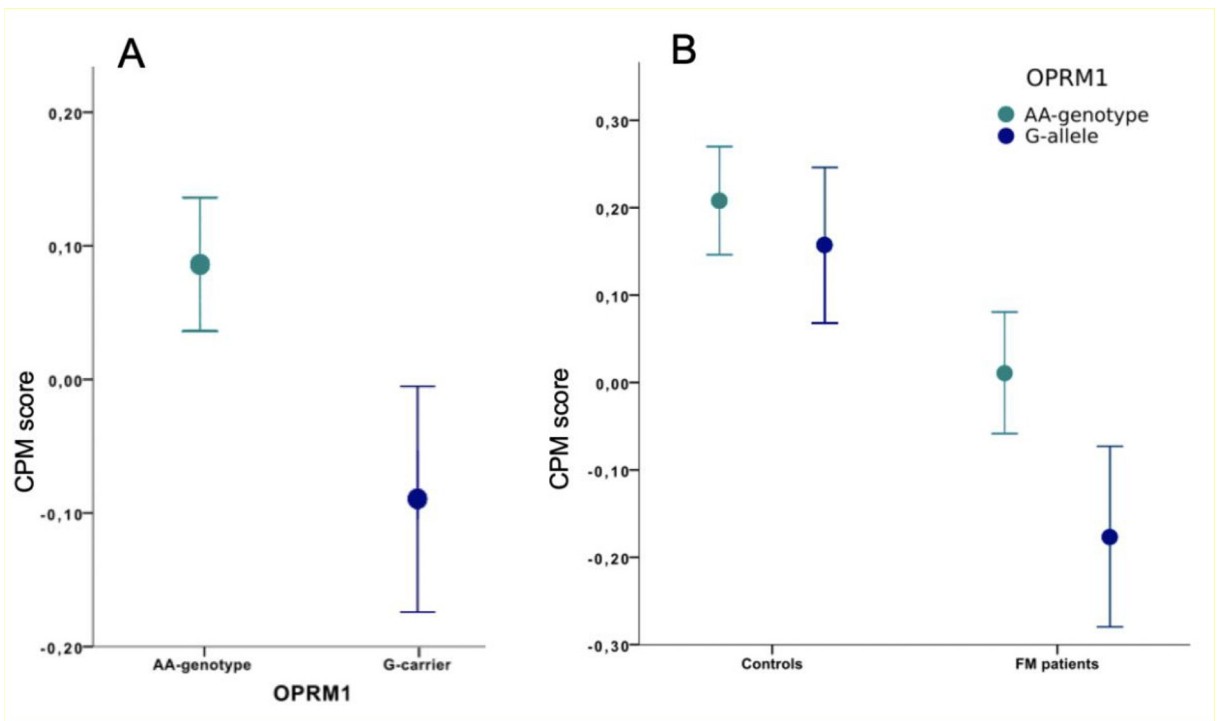

**Fig 2. Conditioned pain modulation (CPM) score based on OPRM1 genotype in the study group.** Fig 2A. Fibromyalgia patients and healthy controls pooled together split by OPRM1-AA versus G-genotype. Results showed significantly reduced CPM score in OPRM1 G-allele carriers (p = 0.019). Fig 2B. CPM score for HC and FM patients separately split by OPRM1-AA vs G-allele. A significant difference was seen for FM patients (p = 0.045) but not for HC. Notable, FM patients with the OPRM1 G-allele had the lowest CPM score among all four groups (indicating less efficient CPM).

### Associations of CPM score with the gene OPRM1

As hypothesized, no statistically significant interactions were found between group (FM, HC) and genetic polymorphisms regarding the CPM score, indicating similar genetic effects regardless of baseline pain level. When both groups were analyzed jointly, a statistically significant effect of OPRM1 (df = 1, F = 5.7, p = 0.019) was found for the CPM score. More specifically, individuals with OPRM1 AA-genotype had higher CPM scores compared to the G-genotype (mean CPM scores: OPRM1 AA 0.085 (SD 0.49), G-genotype -0.093 (SD 0.45), which indicates reduced ability in individuals with OPRM1 G-genotype to activate descending pain inhibition (Fig 2A). Post hoc testing of each group separately revealed significant effects of the OPRM1 genotype on CPM score in FM patients, (df = 1, F = 4.17, p = 0.045), with reduced CPM score in FM patients carrying the OPRM1 G-genotype (mean CPM scores: OPRM1 AA 0.011 (SD 0.54), G-genotype -0.18 (SD 0.47), while no significant effect was seen in HC (Fig 2B). Regarding the serotonergic genes, no significant effects on CPM score were found for 5-HT1a or 5-HTT, respectively.

### Gene-to-gene interactions

There was a gene-to-gene interaction between OPRM1 and 5-HT1a (df = 1, F = 5.38, p = 0.022). Post hoc analyses revealed that individuals with 5-HT1a CC genotype had better CPM score if they had the OPRM1 AA genotype compared with the OPRM1 G-genotype (p = 0.029), indicating better central pain inhibition with the genetic setup of 5-HT1a CC-genotype and OPRM1 AA-genotype (Fig 3). Also, there was a trend that individuals with

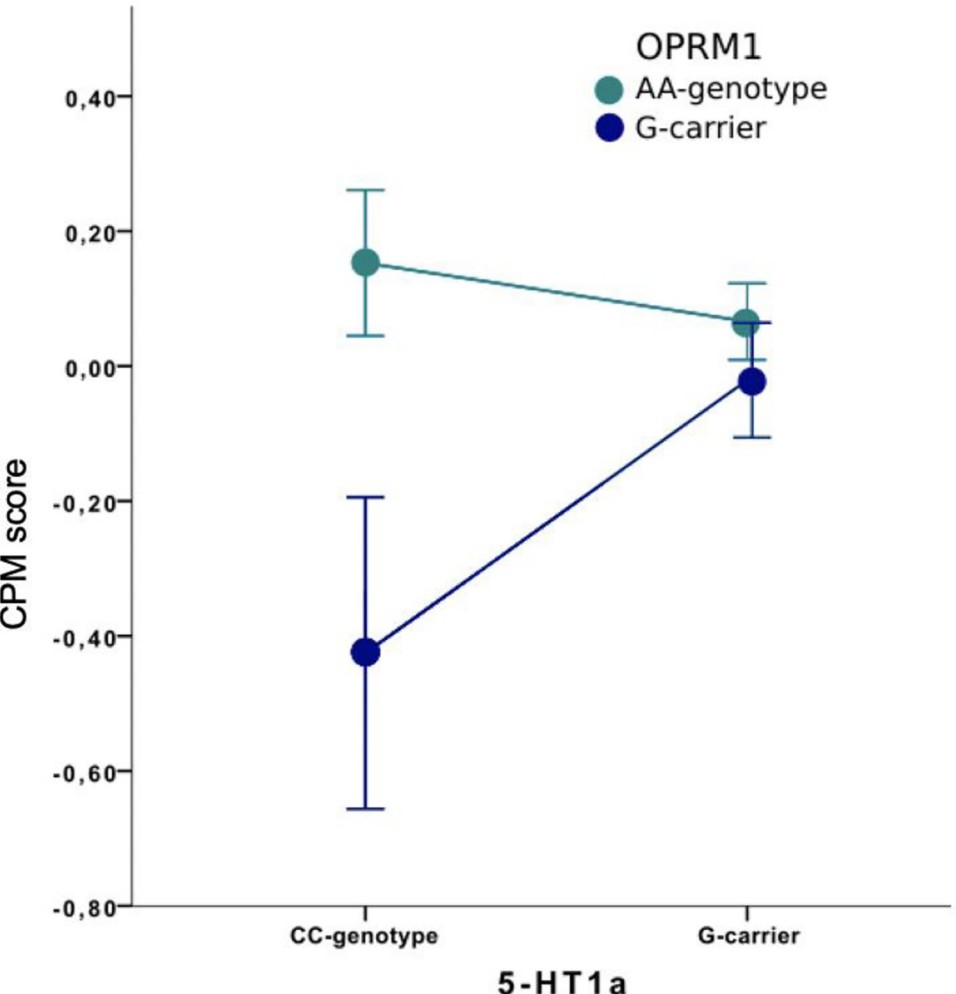

**Fig 3. Genetic interaction of OPRM1 and 5-HT1a.** A significant gene-to-gene interaction between OPRM1 and 5-HT1a was found on conditioned pain modulation (CPM) in fibromyalgia (FM) and healthy controls. Subjects with 5-HT1a CC-genotype had a significantly higher CPM score if they also had the OPRM1 AA-genotype compared to G-genotype (p < 0.05). In accordance, there was a trend in OPRM1 G-carriers where 5-HT1a G-carriers had a higher CPM score than CC-carriers (p = 0.065). PPT = pressure pain threshold. CPM score = (PPT end–PPT baseline)/PPT baseline.

OPRM1 G-genotype and 5-HT1a G-genotype had higher CPM score compared with the 5-HT1a CC-genotype (CPM score CC: -0.43 (SD 0.52) and G -0.02 (SD 0.41); p = 0.065). Regarding the serotonergic genes, no significant interactions were found for 5-HT1a x 5-HTT nor for 5-HTT x OPRM1, and thus, no post hoc analyses were performed.

## Genetic effects on the temporal aspects of CPM

Regarding the effect of the genetic polymorphism in OPRM1 on normalized PPTs assessed over time there was a statistically significant effect of GENOTYPE (OPRM1 AA or G-genotype) (df = 1, F = 4.59, p = 0.034) and a significant GENOTYPE x TIME interaction (df = 3.11, F = 2.93, p = 0.032). Post-hoc analyses reveal that individuals with the OPRM1 G-genotype, compared to the AA-genotype, had significantly lower CPM at start (p = 0.035) as well as after the conditioning stimulus (p = 0.008), revealing a reduced ability to activate descending pain

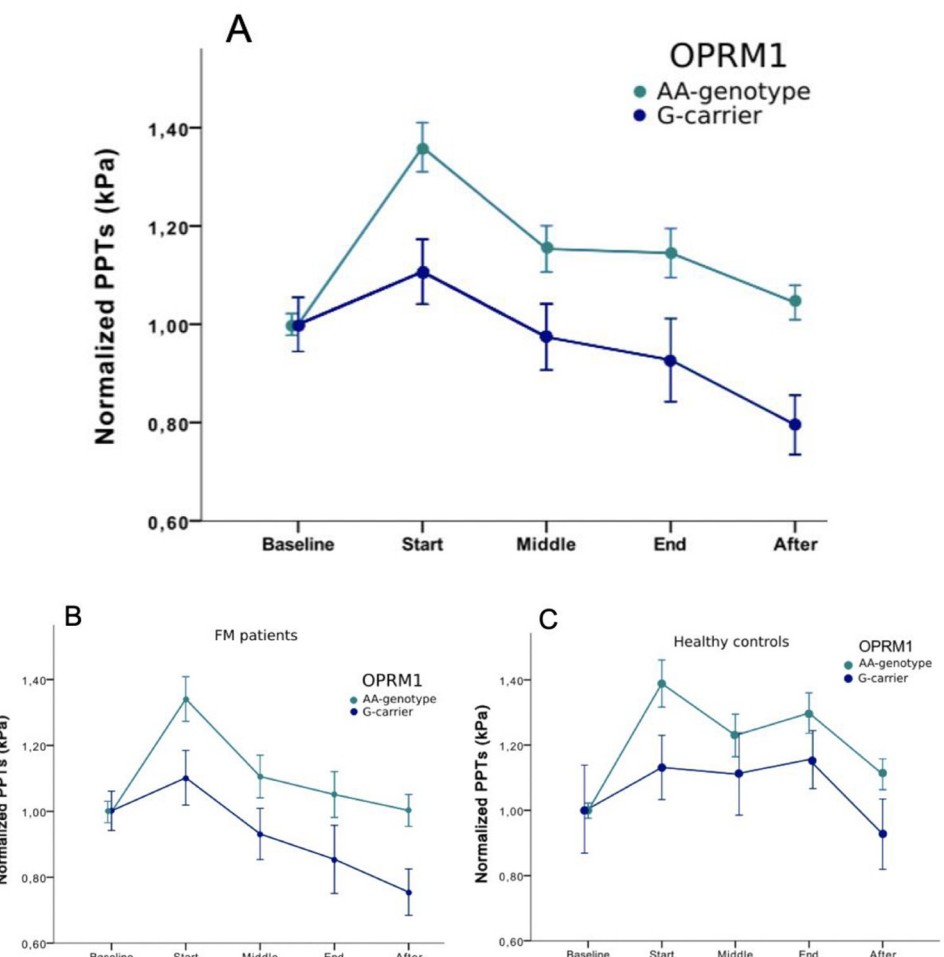

**Fig 4. A.** Normalized pressure pain thresholds (PPTs) (mean +/- SEM) at baseline, start, middle, end and 5 minutes after a standardized assessment for conditioned pain modulation for the whole group divided into OPRM1 AA versus G-genotypes. Individuals with OPRM1 G-genotype had significantly reduced PPTs at start (p = 0.035) and after (p = 0.008). **4B and 4C.** The effect of OPRM1 normalized PPTs over time in fibromyalgia (FM) patients and healthy controls analyzed separately. No significant GENOTYPE x GROUP interaction was found, indicating similar effects in both groups regardless of baseline pain level.

inhibition, with the effect persisting up to 5 minutes after the assessment. The data visualized in Fig 4 indicate that the effect of reduced CPM in OPRM1 G-genotype individuals is similar in FM patients and HC (Fig 4B and 4C). However, post hoc tests failed to show significance when FM and HC were analyzed separately, as power is reduced.

## Assessment of temporal aspects of CPM in FM patients compared to HC

Differences in normalized PPTs over time (baseline, start, mid, end and after 5 minutes) between FM patients and HC were assessed by repeated measures ANOVA. A significant TIME x GROUP interaction (df = 3.16, F = 3.48, p = 0.014) was found. Post-hoc analysis revealed a statistically significant group difference at the end of the CPM paradigm (p < 0.001), indicating better functioning pain inhibitory processes in HC compared to FM patients (Fig 5). Moreover, FM patients had reduced pain sensitivity after 5 minutes with a group mean below the mean PPT at start, indicating pain facilitation after the CPM

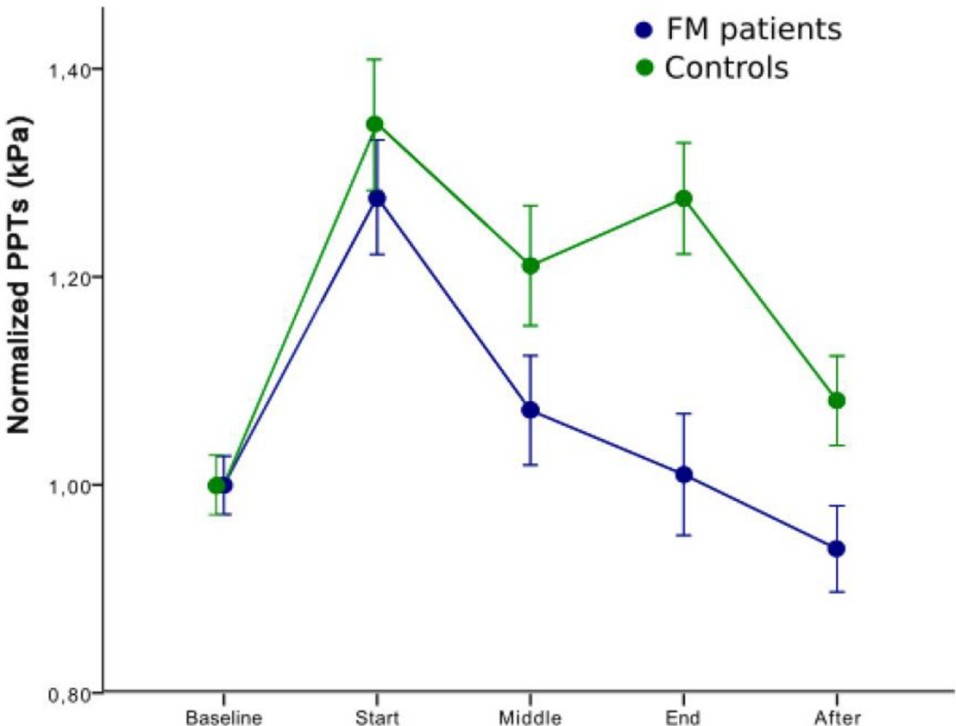

**Fig 5. Normalized pressure pain thresholds (PPTs) (mean ± SEM) at start, middle, end and 5 minutes after a standardized tourniquet test.** The conditioning pain was applied in the upper left arm and PPTs were assessed at the contralateral *m. quadriceps* in FM patients and healthy controls. There was a statistically significant difference between the groups at end (p<0.001) and after (p = 0.041) the assessment.

assessment. This represents a statistically significant difference between the groups, where the HC group still show activated pain inhibition (p = 0.041). As there was a significant group difference in the length of the CPM assessment, an additional model was used to control for this. However, no significant influence of the length of the CPM assessment was found (time x group df = 3.32, F = 3.70, p = 0,009 and time x length of CPM df = 3.32, F = 1.43 p = 0.23).

## Associations between genetic polymorphisms and FM symptoms

There were no statistically significant effects of the assessed polymorphisms of OPRM1 or 5-HT1a regarding symptom severity (FIQ, HAD-A, HAD-D, SF36-BP) or pain sensitivity (average PPT). There was a significant effect of 5-HTT and an interaction between 5-HTT x 5-HT1a regarding HAD-D, i.e. assessment of depression, data reported elsewhere [47]. Also, a significant gene-to-gene interaction with HAD-D was found regarding 5-HTT x 5-HT1a x OPRM1 (F = 6.07, df = 1, p = 0.016).

## Discussion

Here we examined the role of specific functional genetic polymorphisms acting on opioid and serotonergic signaling on CPM in healthy subjects and patients with FM. This is the first study to show that, by inferring from genotype, phenotypes of endogenous opioid signaling significantly influences descending pain modulation in humans as measured by CPM. More specifically, OPRM1 G-allele, mechanistically proposed to increase the affinity for β-endorphins [23], while reducing MOR expression and signaling [24] was associated with decreased CPM, i.e. a reduced ability to activate descending pain inhibition. Furthermore, there was a gene-to-gene

interaction between the OPRM1 and the 5-HT1a gene, revealing that the negative effects of the OPRM1 G-genotype on CPM were seen specifically in individuals with the 5-HT1a CC-geno-type, which associated with increased 5-HT signaling mediated via the 5-HT1a receptor [48]. This interaction is in line with our hypothesis based on our previous report of genetic influ-ence on pain modulation, where we showed gene interactions with a similar pattern during EIH, with carriers of the 5-HT1a CC/OPRM1 G-genotype exhibiting the lowest EIH [11]. Lastly, similar to the results from the previous EIH study, there were no differences between FM patients and HC regarding the effects of the genetic variants on pain modulatory pro-cesses. Thus, these results seem to reflect more general mechanisms influencing pain modula-tory processes and not the underlying dysfunction of CPM in FM.

## The effect of the MOR and the OPRM1 gene on CPM

The opioid system, and μ-opioid receptors in particular, is known to be involved in descending pain modulation. Here, we show that a genetic variant in the OPRM1 gene, coding for the μ-opioid receptor (MOR), independently influences the function of CPM. Individuals with the OPRM1 G-genotype had significantly poorer ability to activate descending pain modulation where endogenous opioids are engaged. In the literature, this genetic variant is proposed to mediate stronger binding of endogenous opioids to the μ-opioid receptor [23], suggesting an increase of function. However, it is well established that individuals with this variant require higher postoperative exogenous opioids for pain relief, instead suggesting a loss of function [25,49]. A study of opioid-naive healthy subjects showed that strong μ-opioid agonists potenti-ated CPM [50], while the opposite was seen in long-term opioid drug users who displayed less efficient CPM, suggesting a dampening effect on the endogenous pain modulatory system by longstanding exposure [51]. On a similar note, the OPRM1 G-allele has been proposed to exhaust the opioid system of a compensatory reaction to chronic exposure to opioid drugs [52] and could thus hypothetically be relevant for the shift from anti- to pronociceptive pain modulatory processes during long-term opioid exposure. Based on this, we hypothesize that the OPRM1 G-genotype plays a role in interindividual differences of endogenous opioid effect on CPM activation. Pecina et al demonstrated that carriers of the OPRM1 G-allele had reduced μ-opioid receptor availability in brain areas implicated in regulation of pain and affect [27]. Thus, the OPRM1 G-allele, favoring a higher MOR affinity for β-endorphins [23], could hypothetically have a negative impact on opioid signaling, due to lower availability of MOR. During the CPM assessment in the present study, a strong painful sensation was evoked by the conditioning stimulus (mean rating of worst pain was VAS 77/100) which most likely induced an immediate engagement of central endogenous opioids. Hypothetically, individuals with the OPRM1 G-allele would have a stronger initial binding of endogenous opioids to MOR leaving less receptors available to newly activate and initiate pain inhibition from antinociceptive neu-rons downstream, resulting in a loss of function of pain inhibition as reported here.

Compared to HC, FM patients have a reduced MOR binding capacity in brain regions implicated in pain regulation, including the opioid rich rostral anterior cingulate cortex (rACC) [53,54]. Importantly, lower MOR binding capacity was associated with reduced pain related activation of rACC in FM patients [55]. Moreover, FM patients display less functional connectivity between the rACC and other parts of the brain's pain inhibitory network [56] as well as decreased cortical thickness of rACC, related to a longer duration of FM [57]. In anal-ogy with our hypothesis regarding the OPRM1 polymorphism, Schrepf et al. (2016) proposed that initially high levels of endogenous opioids in FM patients [16] would cause downregula-tion and/or reduced affinity of MOR in the antinociceptive brain regions, thus explaining the failure to activate descending inhibitory mechanisms during pain stimulation in FM. In this

scenario, FM patients with the OPRM1 G-genotype would be expected to have a further reduction of MOR signaling during evoked pain, resulting in even less efficient descending pain inhibition. Indeed, FM patients in our study had reduced CPM compared to HC, as has been repeatedly reported previously [12,58], interestingly with the lowest CPM scores seen in FM patients carrying the OPRM1 G-allele (Fig 2B). Thus, the effect of the OPRM1 G-allele variant seems to exist regardless of baseline function of descending pain inhibition or the presence of a chronic pain condition.

### Interaction between opioid and serotonergic genes on CPM

The opioid system interacts with the serotonergic pain regulatory system in an antagonistic, time- and state dependent way [59]. However, the specific effect of 5-HT1a on opioid mediated pain modulation is difficult to interpret. Stimulation of MOR give a first order analgesic effect, but can have a second order hyperalgesic effect, i.e. opioid-induced hyperalgesia, which in turn can be counteracted by stimulation with 5-HT1a agonists [18]. In fact, 5-HT1a agonists have the opposite profile to opioids, i.e., a first order pronociceptive, followed by a second order analgesic effect [60] and the 5-HT1a receptor is shown to mediate inhibition of pain [61]. Thus, the effects of 5-HT signaling on pain perception seem to depend on the state of the opioid system.

On a genetic level, the proposed mechanism of the 5-HT1a G-genotype is that it leads to a reduction in 5-HT transmission, as it yields upregulation of presynaptic inhibitory 5-HT1a auto-receptors in the raphe nuclei where 5-HT is synthesized, and downregulation of postsynaptic 5-HT1a receptors [48,62]. In this study, we found significant gene-to-gene interactions between genes coding for the 5-HT1a receptor and MOR. More specifically, individuals with 5-HT1a CC/OPRM1 G-genotype had significantly reduced CPM score, i.e. reduced ability to activate descending pain modulation, as compared to other genetic combinations. In fact, these two genotypes combined produced negative CPM scores, meaning pain facilitation during CPM. As described above, the OPRM1 G-genotype was in itself associated with a lower CPM score, but this effect was driven by individuals with genetically inferred increased 5-HT signaling and higher concentrations of postsynaptic 5-HT1a receptors. The latter is in accordance with the interactions between 5-HT1a agonists and opioids [18]. Furthermore, these data are in line with our previous report on these genes influence on EIH [11]. Interestingly, both methodologies, i.e. CPM and EIH, produced gene-to-gene interactions in the same direction based on proposed mechanisms of the opioid and 5-HT1a polymorphisms studied, and no difference was found based on baseline pain level (FM patients or HC).

Contrary to our findings during EIH [11], no statistically significant interactions between OPRM1 and 5-HTT affecting CPM were found. Previous studies regarding the effects of the 5-HTT polymorphism on CPM have shown inconsistent results. Healthy carriers of the low expressing 5-HTT polymorphism had reduced CPM affecting both mechanical- and heat pain stimuli when the tourniquet was used as conditioning stimulus [14], reduced CPM during cold pressor test as measured by non-painful heat stimuli but not painful heat stimuli [63], as well as normal CPM assessed by tonic heat stimuli during a cold pressor test, the latter was also seen in FM patients [58]. The inconsistent effects of the 5-HTT polymorphism on CPM may explain why we failed to reveal a 5-HTT and OPRM1 interaction in the current study.

### Limitations

The main limitation of our study is that the effects of opioid and serotonergic signaling are inferred from genotypes in opioid and serotonergic genes. Therefore, we should not draw definite conclusions to underlying biological mechanisms. Nevertheless, genetic variability

influences pain phenotypes [64]. This method is well established in the literature and allows us to study pain regulation without pharmacological manipulation, which has an additive value to current knowledge of pain regulatory processes. Furthermore, as the analyses of the FM and HC groups separately are based on rather low sample sizes for the uncommon genotypes of the investigated genetic polymorphisms, the results must be regarded as exploratory and need to be reproduced in larger cohorts.

## Conclusions

This is the first study to show that the gene OPRM1, coding for a main structure of endogenous opioid signaling, independently influences pain behavior without exogenous opioids being involved. More specifically, a polymorphism known to alter the expression of, and binding to, the μ-opioid receptor influences an individual's ability to activate descending pain inhibition during ischemic pain. Furthermore, a gene-to-gene interaction was found showing that the OPRM1 G-genotype/5-HT1a CC- genotype conferred significantly reduced ability to activate descending pain modulation, as compared to other genetic combinations. The results revealed that the effect of the OPRM1 G-genotype on CPM was driven by individuals with genetically inferred higher 5-HT signaling and higher concentrations of postsynaptic 5-HT1a receptors. Interestingly, this gene-to-gene interaction is similar to our previous results from a study assessing EIH in HC and FM patients, thus validating the findings to, not only two different methodologies to activate descending pain inhibition, but also two different cohorts.

## Acknowledgments

The authors thank the personel at KIGene, Karolinska Institutet, Stockholm, Sweden for excellent collaboration and for performing the DNA extraction and genotyping, we also thank for the excellent patient recruitment and administrative support during this project.

## Author Contributions

**Conceptualization:** Eva Kosek.

**Data curation:** Jeanette Tour, Angelica Sandström, Diana Kadetoff, Martin Schalling.

**Formal analysis:** Jeanette Tour.

**Funding acquisition:** Eva Kosek.

**Investigation:** Jeanette Tour.

**Methodology:** Jeanette Tour, Martin Schalling, Eva Kosek.

**Project administration:** Martin Schalling, Eva Kosek.

**Resources:** Martin Schalling, Eva Kosek.

**Supervision:** Eva Kosek.

**Writing – original draft:** Jeanette Tour, Eva Kosek.

**Writing – review & editing:** Jeanette Tour, Angelica Sandström, Diana Kadetoff, Martin Schalling, Eva Kosek.

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
