## [Decision Letter · Decision Letter 0]

8 Sep 2022

PONE-D-22-13588The OPRM1 gene and interactions with the 5-HT1a gene regulate conditioned pain modulation in fibromyalgia patients and healthy controlsPLOS ONE

Dear Dr. Tour,

Thank you for submitting your manuscript to PLOS ONE. After careful consideration, we feel that it has merit but does not fully meet PLOS ONE’s publication criteria as it currently stands. Therefore, we invite you to submit a revised version of the manuscript that addresses the points raised during the review process.

First of all, I would like to apologize for the length of the review process. Unfortunately, one reviewer initially agreed, but was then unable to submit his report, despite repeated reminders. I had to look for a new second reviewer. After receiving his report, I was faced with the dilemma of having two extremely different recommendations. That's why I was looking for a third reviewer, but so far without success. In order not to lengthen the review period even more, I have now decided on the judgment "Major Revision". I would like to ask you to pay particular attention to the criticisms of reviewer #2.

We look forward to receiving your revised manuscript.

Kind regards,

Wolfgang Blenau

Academic Editor

PLOS ONE

Journal Requirements:

Reviewers' comments:

Reviewer's Responses to Questions

**Comments to the Author**

1. Is the manuscript technically sound, and do the data support the conclusions?

Reviewer #1: Yes

Reviewer #2: Partly

2. Has the statistical analysis been performed appropriately and rigorously? 

Reviewer #1: Yes

Reviewer #2: No

3. Have the authors made all data underlying the findings in their manuscript fully available?

Reviewer #1: Yes

Reviewer #2: Yes

4. Is the manuscript presented in an intelligible fashion and written in standard English?

Reviewer #1: Yes

Reviewer #2: Yes

5. Review Comments to the Author

Reviewer #1: This is an interesting manuscript showing that the OPRM1 G-allele was associated with decreased CPM, as well as interactions between the OPRM1 and the 5-HT1a gene, with reduced CPM scores in individuals with the OPRM1 G*/5-HT1a CC genotype, regardless of pain phenotype. The manuscript is well-written and complements well the existing literature. This reviewer has no suggestions for further improving the manuscript.

Reviewer #2: This is a very interesting study with potentially important implications. The experimental design is excellent and well-described. The statistical analyses are appropriate and considered the nature of repeated testing and potential pain ceiling effects for those participants who had to stop the procedure early. The hypotheses are grounded in the existing literature.

Despite these strengths, there appears to be a very significant issue relating to sample size. Because of the nature of the genotype analyses, we would expect that there would be very small numbers of participants in the OPRM1 G-carrier and 5-HT1a CC-genotype groups. Unfortunately, the authors do not present these numbers but it seems likely that there would be fewer than 10 subjects and therefore far fewer than would be ideal to evaluate the effects of interest. This is unfortunate particularly because the manuscript is framed around genotype interactions. After evaluating the study plan for the parent study, it is does not appear that additional participants are likely to be recruited, allowing for appropriate statistical inference. Therefore this material is better suited as a very exploratory analysis with much less definitive conclusions.

6. PLOS authors have the option to publish the peer review history of their article (what does this mean?). If published, this will include your full peer review and any attached files.

Reviewer #1: No

Reviewer #2: No

---

## [Author Response · Author response to Decision Letter 0]

24 Oct 2022

Reviewer # 1: Answer: We thank the reviewer for his/her kind words. We are happy that the reviewer found our manuscript interesting, well written and without need for further improvement. 

Reviewer #2: This is a very interesting study with potentially important implications. The experimental design is excellent and well-described. The statistical analyses are appropriate and considered the nature of repeated testing and potential pain ceiling effects for those participants who had to stop the procedure early. The hypotheses are grounded in the existing literature. Despite these strengths, there appears to be a very significant issue relating to sample size. Because of the nature of the genotype analyses, we would expect that there would be very small numbers of participants in the OPRM1 G-carrier and 5-HT1a CC-genotype groups. Unfortunately, the authors do not present these numbers but it seems likely that there would be fewer than 10 subjects and therefore far fewer than would be ideal to evaluate the effects of interest. This is unfortunate particularly because the manuscript is framed around genotype interactions. After evaluating the study plan for the parent study, it is does not appear that additional participants are likely to be recruited, allowing for appropriate statistical inference. Therefore, this material is better suited as a very exploratory analysis with much less definitive conclusions.

Answer: 

We thank the reviewer for positive comments of an adequate statistical method and a grounded hypothesis raising this question. 

We also thank the reviewer for the relevant comment on sample size. We present the sample sizes with corresponding genotype in Table 2, however they are presented in groups of FM and HC. For our main analyses the groups are pooled, and the exact sample size for the genotypes the reviewer mentions are OPRM1 G-carrier N=28 and for 5-HT1a CC- genotype N=26, i.e. not fewer than 10 individuals. To clarify this in the manuscript, we have added columns to Table 2 for the sample size of the whole group. We believe our sample size is statistically sufficient for making inferences. Yet, we fully agree with the reviewer that the study findings would be more robust with more individuals recruited, however, we are unable to recruit more participants. Regarding the analysis where FM and HC are analyzed separately, the reviewer is correct in that sample size is low and should be inferred as exploratory. We have added a comment in the discussion that this specific analysis is exploratory and needs to be replicated in a larger cohort. Again, we thank for the comment as it has improved our manuscript substantially. 

We hope that we have correctly understood the reviewer’s question and provided a satisfactory answer.

---

## [Editor Report · Decision Letter 1]

27 Oct 2022

The OPRM1 gene and interactions with the 5-HT1a gene regulate conditioned pain modulation in fibromyalgia patients and healthy controls

PONE-D-22-13588R1

Dear Dr. Tour,

We’re pleased to inform you that your manuscript has been judged scientifically suitable for publication and will be formally accepted for publication once it meets all outstanding technical requirements.

Kind regards,

Wolfgang Blenau

Academic Editor

PLOS ONE